# Comparison of Diagnostic Accuracy of Physical Examination and MRI in the Most Common Knee Injuries

**Przemysław Krakowski** [1,*] **, Adam Nogalski** [2] **, Andrzej Jurkiewicz** [1] **, Robert Karpiński** [3] **, Ryszard Maciejewski** [4] **and Józef Jonak** [3]

[1]  Orthopaedic Department, Łęczna Hospital, Krasnystawska 52 str, 21-010 Łęczna, Poland; jurkiewicz16@wp.pl

[2]  Department of Trauma Surgery and Emergency Medicine, Medical University of Lublin, Staszica 16 str, 20-081 Lublin, Poland; adamnogalski5@gmail.com

[3]  Lublin University of Technology, Faculty of Mechanical Engineering, Department of Machine Design and Mechatronics, Nadbystrzycka 36, 20-618 Lublin, Poland; r.karpinski@pollub.pl (R.K.); j.jonak@pollub.pl (J.J.)

[4]  Department of Human Anatomy, Medical University of Lublin, Jaczewskiego 4 str, 20-090 Lublin, Poland; ryszard.maciejewski@umlub.pl

*  Correspondence: Przemyslaw.krakowski84@gmail.com; Tel.: +48-606555183

**Abstract:** Purpose: This study evaluated the diagnostic accuracy of physical examination and magnetic resonance imaging (MRI) in knee injuries. Methods: Ninety-six patients at a regional hospital were included in the study. Each participant underwent a physical examination in which menisci and ACL were evaluated. Knee joint MRI was collected from each patient. Physical examination and MRI scans were then compared with knee arthroscopy findings as a golden standard for meniscal and ligamentous lesions. The data were analyzed and specificity and sensitivity were calculated and correlated on receiver operating characteristics (ROC) curves. Results: Knee arthroscopy diagnosed 32 total ACL ruptures, 45 medial meniscus and 17 lateral meniscus lesions. Three patients were diagnosed with bilateral meniscal lesions. The highest sensitivities were the McMurray test (87.5%) for medial meniscus (MM) and the Thessaly test (70%) for lateral meniscus (LM). The most sensitive ACL test was Lachman (84.5%), whereas, the pivot shift and Lelli tests were the most specific (98.5%). MRI was highly sensitive for MM (96%) with specificity of 52%. MRI showed lower sensitivity (70%) and higher specificity (85.5%) for LM. The specificity of MRI for ACL rupture was 92%, with sensitivity only 75%. Conclusion: McMurray and Apley tests for meniscal lesions seem the most appropriate in daily practice. A combination of lever signs, pivot shifts (PSs) and Lachman tests showed the best sensitivity and specificity in detecting ACL deficiency, and was superior to MRI.

**Keywords:** physical examination; MRI; knee; meniscus; ACL; knee injury

## 1. Introduction

The knee joint is one of the most common joints subjected to injuries. Moreover, due to its localization and function, it is of vital importance to working ability, daily tasks and recreational and professional sports. Damage to tissues such as menisci, ligaments or hyaline cartilage can lead to irreversible osteoarthritic changes of the joint [1,2]. Therefore, quick and accurate diagnosis of intraarticular lesions, which is necessary for selecting adequate treatment, is of great importance [3].

An abundance of mechanisms can lead to knee joint failure, which is one of the factors that makes diagnosis difficult [4–6]. Delay in introducing proper treatment has a great effect on hyaline cartilage

and leads to irreversible changes in its structure [7,8]. Therefore, apart from a thorough medical history, specific diagnostic tests must be implemented, and even then it is not uncommon for even very experienced medical professionals to have difficulties making an adequate diagnosis.

Over the last few decades, magnetic resonance imaging has been continuously gaining acceptance among the orthopaedic society as a diagnostic tool, ousting the use of arthroscopy as a diagnostic modality [9]. However, with the increasing popularity and accessibility of MRI, the amount of unnecessary MRI increases, having reached nearly 40% of all examinations [10]. Moreover, the study performed by Solivietti et al. showed that nearly 20% of patients have not had an appropriate physical examination prior to their MRI [10]. Highly sophisticated imaging modalities such as MRI are expensive and have a long waiting list; therefore, it seems advisable that recommendations for MRI should be justified by clinical appearance. On the contrary, the use of MRI can decrease the number of unnecessary surgical treatments [11]. In this study, over 30% of the patients waiting for surgery after MRI evaluation of the affected knee were disqualified from surgical treatment due to benign MRI findings. Although arthroscopy is considered to be a safe surgical intervention with low incidence of adverse effects [12], one should not forget that it is a surgical procedure with potentially hazardous side effects.

Given the information provided above, it seems important to work out a physical examination algorithm that would enable reproducible, fast and accurate knee examinations for orthopaedic surgeons. This would facilitate fast and accurate diagnosis and adequate treatment, which could prevent further damage to the knee joint and the onset of osteoarthritic changes in cartilage.

The aim of this study was to evaluate the sensitivity and specificity of the most common tests used in diagnosing meniscal and anterior cruciate ligament (ACL) injuries and to compare the tests with each other. Moreover, the accuracy of physical examinations of common knee injuries was contrasted with the use of MRI. This study could help in in creating a physical examination algorithm.

## 2. Materials and Methods

This study was conducted in the Orthopaedic Department of Łęczna hospital in Poland between February 2014 and June 2017. Each patient signed written consent at the moment of admission to the hospital that utilized medical data for treatment and clinical applications. The hospital holds all the necessary approvals and accreditations related to patient data processing in the treatment and research fields. Prior to being admitted to the Orthopaedics and Traumatology Department, each patient signed their consent of the processing of personal and medical data. The data acquisition had no impact on the diagnostic process or further treatment, and therefore approval from ethics committee was not obligatory for the study.

### 2.1. Participants

Total of 96 patients that had previously qualified for knee arthroscopy were included in the study. The study group consisted of 49 females and 47 males, with 52% of the patients treated due to dominant extremity involvement. The mean age of patients seeking medical attention in this study was 45 years. The mean time between the onset of symptoms and treatment was 44 months, and the patients with a gradual onset of symptoms but without a history of injury waited the longest for surgery, with mean time of 71 months. Over 60% of patients prior to surgical treatment had undergone some kind of conservative treatment such as physiotherapy, viscosupplementation or steroid injections. The most common form of conservative treatment was physiotherapy, which was prescribed to 58 patients prior to surgery. The effects of conservative treatment were not evaluated in this study, and the analysis was carried out only to determine whether the conservative treatment postponed the surgery itself. The clinical and baseline characteristics of the participants with summarized diagnoses based on arthroscopic findings are represented in Table 1. The study was approved by Bioethical Committee by Medical University in Lublin with the number of approval KE- 0254/262/2019.

Due to great diversity of patients treated at the clinic, we have established the following exclusion criteria:

- musculo–skeletal diseases that might affect clinical manifestation of symptoms;
- previous fractures of the distal femur or proximal tibia;
- metal implants that may impair the MRI evaluation;
- prior surgical treatment of the knee joint;
- tumors of the knee joint;
- incomplete physical examination due to pain experienced by the patient;
- viscosupplementation, platelet rich plasma (PRP) or steroid injections in the last three months; and
- lack of consent for storing and using medical images.

**Table 1.** Clinical and baseline characteristics.

| Parameter | |
|---|---|
| Age, year (mean ± SD) | 45 ± 16 |
| Sex, n (%) | Females 49 (51%), Males 47 (49%) |
| Duration of symptoms, months | 43.9 ± 70.8 |
| History of trauma, n (%) | 60 (62.5%) |
| Involvement of the dominant extremity, n (%) | 5 (31.25%), 50 (48%) |
| Opposite extremity involvement, n (%) | 14 (14.5%) |
| Physiotherapy prior to surgery, n (%) | 60 (62.5%) |
| Medial meniscus lesions, n (%) | 45 (46.8%) |
| Lateral meniscus lesions, n (%) | 17 (17.7%) |
| Bilateral meniscus lesions, n (%) | 3 (3.125%) |
| ACL tear, n (%) | 30 (31.25%) |
| Isolated ACL tear, n, (% of ACL tears) | 3 (10%) |
| ACL tear and MM lesion, n, (% of ACL tears) | 4 (13.3%) |
| ACL tear and LM lesion, n, (%of ACL tears) | 3 (10%) |
| ACL tear, MM and LM lesion, n, (% of ACL tears) | 0 (0%) |
| ACL tear and isolated chondral defect, n, (% of ACL tears) | 6 (20%) |
| ACL tear, MM lesion and chondral defect, n, (% of ACL tears) | 9 (30%) |
| ACL tear, LM lesion and chondral defect, n, (% of ACL tears) | 5 (16.6%) |
| ACL tear, LM and MM lesion and chondral defect, n, (% of ACL tears) | 2 (6.7%) |

## 2.2. Physical Examination

During admission to the hospital, a detailed medical history was taken and a physical examination was performed for each patient. Each physical examination adhered to identical protocol. MRI was not shown to the examining surgeon at the time.

Afterwards, the patient was stripped to their underwear and their gait was evaluated. Such factors as muscle atrophy, limb alignment and passive and active knee range of motion (ROM) were noted. Every joint was palpated in order to detect tenderness. After a general evaluation, a detailed knee examination with the use of dedicated ACL and meniscal tests was performed. Classical tests used to detect meniscal pathologies were performed, including McMurray [13], Apley [14] and Thessaly [15] tests, along with a palpation of the joint line. The detection of ACL tears was based on four tests including Lachman [16], anterior drawer [17], pivot shift (PS) [18] and lever sign [19]. To the best of our knowledge, this study is one of the first to evaluate the diagnostic accuracy of lever sign in ACL tears. Accuracy of the diagnostic tests was determined by the specificity and sensitivity of each test in detecting meniscal, chondral and ACL lesions. For each test, the number of false negative, false positive, true positive and true negative results was counted. The results were compared with the use of the receiver operating characteristic (ROC) curve.

### 2.3. Magnetic Resonance Imaging of the Knee Joint

All patients included in the study provided MRI images with a radiologist description. All MRI was performed on 1.5 T coils; however, there was no standard protocol of examination. The sequences used in the examination were selected by a supervising radiologist at the site of examination. MRI images were available to the orthopaedic surgeon only after having performed a standardized physical examination, in order not to bias the diagnosis. The evaluation of articular cartilage, menisci and ACL were made by the radiologist at the time of the MRI examination. All cartilage MRI findings were described with the use of the International Cartilage Repair Society (ICRS) grading system [20]. All MRI findings stated in the radiologist report of the examination were then transferred to the study protocol for later analysis. Moreover, it is worth mentioning that the average time between MRI and arthroscopy was 3.6 months (SD = 3.33), which is representative of the actual time that patients await surgery. Such a short time between MRI and arthroscopy should not influence the MRI's diagnostic accuracy.

### 2.4. Knee Arthrsocopy

All arthroscopies were performed according to standard protocol, with the use of a 30 degree scope by an arthroscopy specialized surgeon. The surgical procedures were performed one day after admission to the hospital and physical examination. All tissues were visualized and probed with an arthroscopic hook to determine their consistency. Consecutively, the suprapatellar, lateral and medial pouches were visualized. At the time of evaluating the pouches, the patellofemoral joint was inspected for cartilage damage. Next, the lateral and medial compartments of the knee were visualized, and special care was taken in examining menisci and cartilage. At the end of the evaluation of the joint, the ACL and posterior cruciate ligament (PCL) were evaluated with the use of an arthroscopic hook.

Meniscal tears were classified according to the International Society of Arthroscopy, Knee Surgery, and Orthopedic Sports Medicine (ISAKOS) into longitudinal vertical, bucket handle, horizontal, radial, flap vertical and horizontal and multidirectional [21]. All arthroscopic findings were pointed out in the surgical report then transferred to the study protocol for later analysis.

### 2.5. Data and Statistical Analysis

Statistical analysis of the research results was conducted using the Dell Statistica (data analysis software system), version 13 (2016) and Microsoft Excel 2013 software. To describe the distribution of the responses' basic features, classical statistical measures were used:

- For categorical variables: number and percentages.
- For continuous variables: mean and standard deviation.

Kolmogorov–Smirnov, Lilliefors and Shapiro–Wilk tests were used to determine whether the tested variables had a normal distribution.

To study the relationships between the variables, the following methods were used:

- Influence of variables on physical examination tests: Pearson correlation coefficient r.
- Comparison of logistic regression steps: chi-square test.
- Time passing from MRI to physical examination: classic one-way analysis of variance (ANOVA).

Receiver operating characteristic (ROC) curve analysis was used to compare the diagnostic properties of different methods. For the comparison of the different ROC curves, 95% confidence intervals were used for areas under the ROC curve.

In order to determine the accuracy of the sets of the diagnostic methods, a logistic regression analysis was applied by assessing the prediction based on their actual state, which was evaluated during an arthroscopy of the knee joint. The best set of predictors (diagnostic tests) was sought in this analysis using the reverse selection method. After obtaining the best set of clinical predictors for logistic analysis, the diagnostics of clinical test sets were compared with magnetic resonance

results, introducing an appropriate resonance result. The sensitivity and specificity of each study were calculated on the basis of ROC curves. Statistically significant results were those that were significant at the typical materiality level (i.e., when $p < 0.05$).

## 3. Results

The medial compartment was more prone to meniscal tears than the lateral compartment. In the study group, 65 patients were diagnosed with meniscal tears: 45 with medial meniscus, 17 with lateral meniscus and 3 patients with bilateral meniscal lesions. Meniscal lesions commonly coexisted with chondral lesions.

ACL injuries were diagnosed in 42 patients, out of which 32 suffered from total tear and 10 patients presented subtotal injuries. The ACL injuries significantly correlated with medial meniscus tears ($p = 0.044$), the medial compartment ($p = 0.03$) and PFJ cartilage lesions ($p = 0.006$). There have been no statistically significant differences in regard to sex ($p = 0.1$) or dominant extremity ($p = 0.4$). For the purpose of the study, only total tears of the ACL were correlated with the physical examination.

### 3.1. Meniscal Lesions Diagnostic Accuracy of Physical Examinations and MRI

Combined sensitivity of the meniscal test was 91% for the medial meniscus. The most sensitive test used in the diagnostics of medial meniscus lesions was the McMurray test, which showed a sensitivity of 87.5% with 52% specificity. Comparable sensitivity and specificity were shown in regards to the Thessaly test, 85.4% and 54%, respectively. The diagnostic accuracy of the Thessaly test was shown to be dependent on the range of motion (ROM) of the joint. If incomplete ROM was noted, the accuracy of the examination dropped significantly ($p = 0.012$). Palpation of the joint line showed significantly lower diagnostic accuracy than all other tests in this study ($p = 0.113$). The greatest area under curve on (AUC) on the ROC graph was shown for the McMurray and Thessaly tests. Except for joint palpation, there were no statistically significant differences between all the tests used in the diagnosis of medial meniscus lesions.

In the case of lateral meniscus, the Thessaly test proved to be most sensitive (70%), with specificity reaching 84%. The McMurray test showed a specificity of 83% and sensitivity of 65%. Similar to medial meniscus, joint line palpation in lateral meniscus lesions showed significantly inferior results in regard to diagnostic accuracy among all tests evaluated in this study ($p = 0.348$). The best ratio of sensitivity to specificity for LM based on AUC was shown for the Apley and Thessaly tests.

All tests, both for the medial and lateral meniscus, were independent from coexisting chondral damage, and did not influence the specificity and sensitivity of the tests in a significant way. The results are shown in Table 2 and Figures 1 and 2.

In this study we have noted gross diversity in diagnostic accuracy of MRI depending on the anatomical location of the meniscus lesion. The highest sensitivity was noted for posterior horns of the medial and lateral menisci, 93% and 77%, respectively. However, the sensitivity for detecting meniscal body lesions was only 55% for MM and 53% for LM. MRI showed the lowest sensitivity in imaging anterior horn tears of the MM (20%) and LM (33%). In order to determine the value of MRI in detecting meniscal pathologies, the menisci were further treated as a whole organ, and if MRI detected any lesion of menisci at any anatomical location it was treated as a positive result. The overall sensitivity of MRI was estimated to be 96% for MM and 70% for LM. The specificity was higher for LM (at 85.5%) than MM (at only 52%). The results were correlated with logistic regression for all meniscal tests in order to determine the value of MRI examination in detecting meniscal lesions over physical examination. The overall physical examination sensitivity (91%) was lower than MRI (95.8%) in detecting medial meniscus lesions; however, the results were not statistically significant. There was also no difference in diagnostic accuracy between logistic regression for physical examinations and MRI in detecting lateral meniscus lesions. The results are shown in Table 2 and Figures 3 and 4.

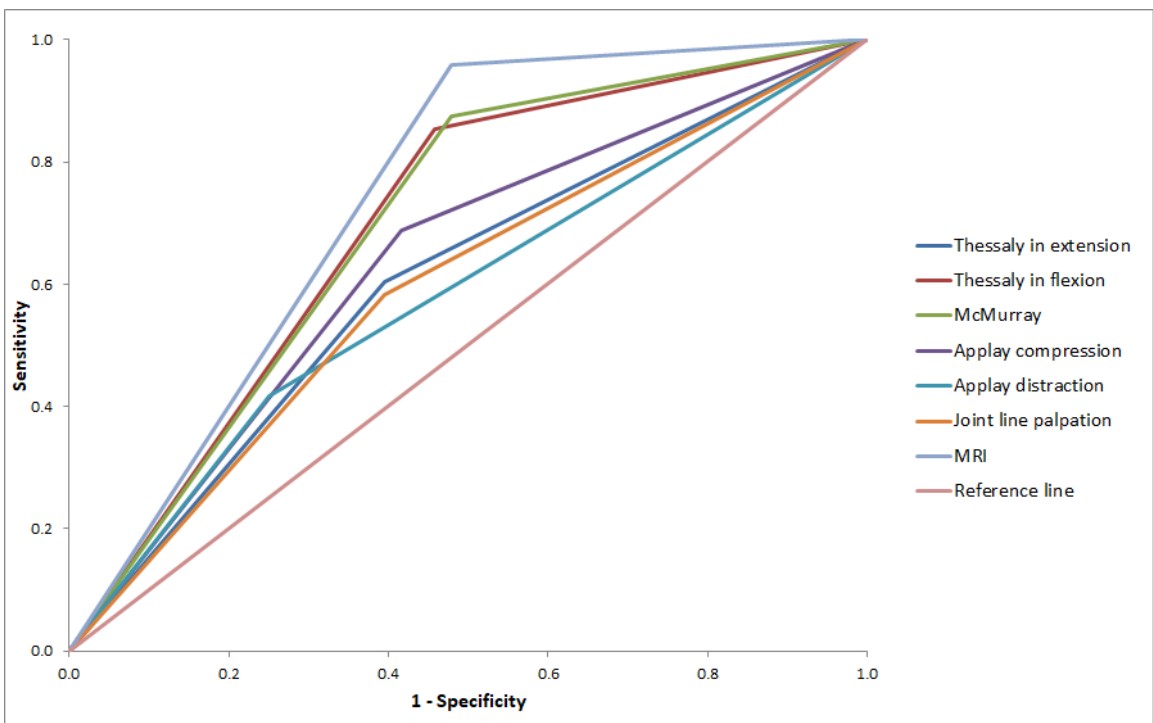

**Figure 1.** ROC curve describing the relation of sensitivity to specificity in the diagnostic tests for the medial meniscus.

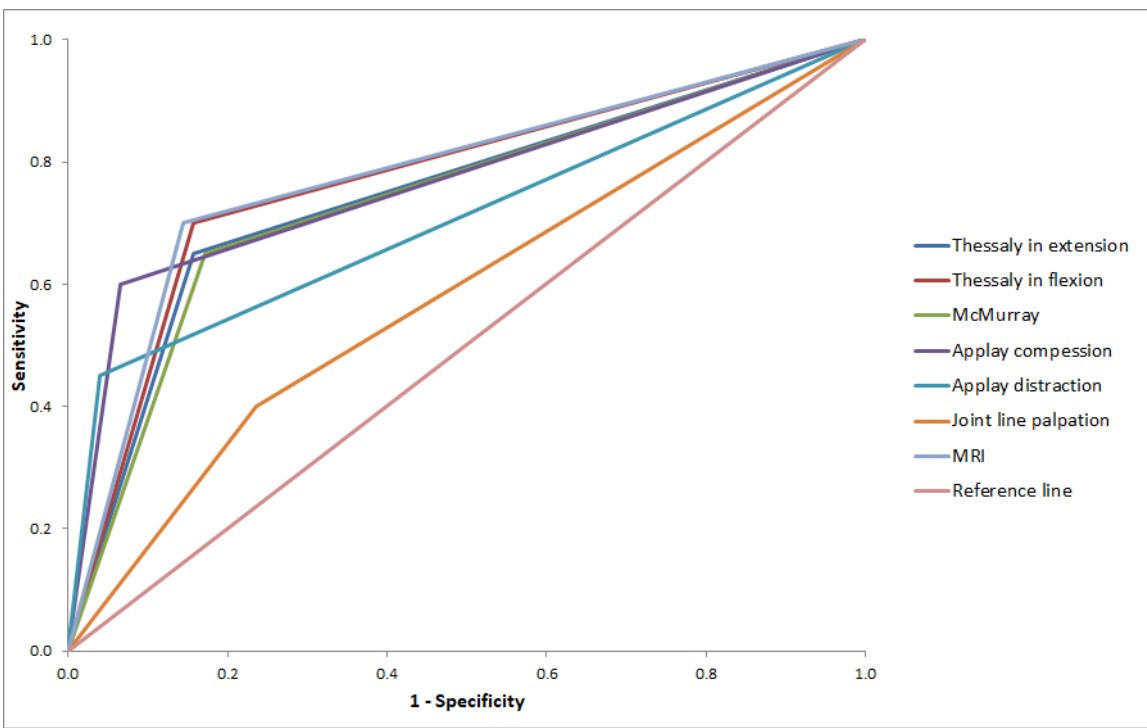

**Figure 2.** ROC curve describing relation of sensitivity to specificity in the diagnostic tests for lateral meniscus.

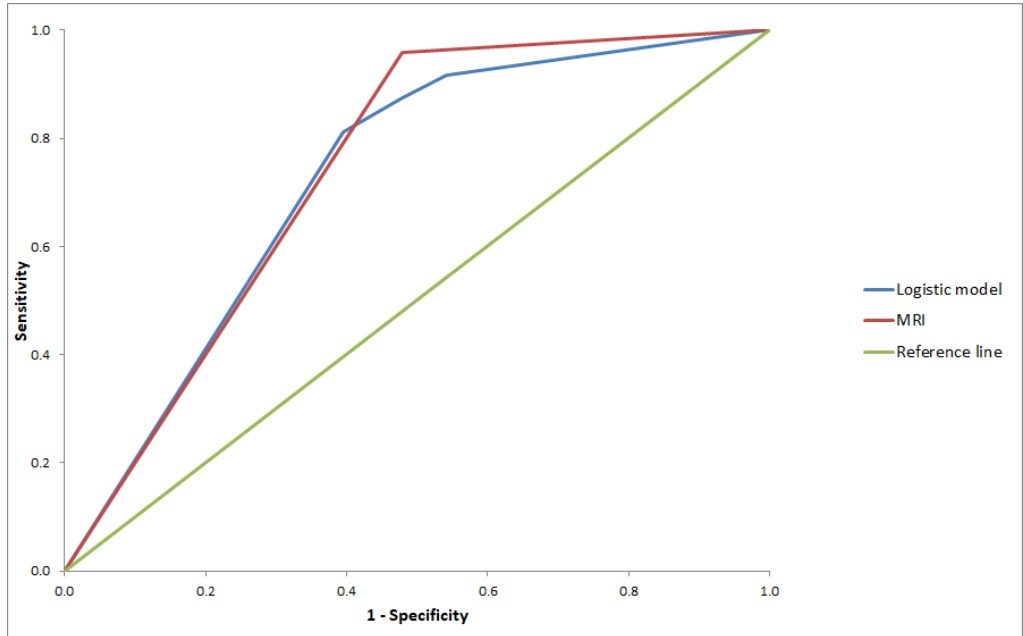

**Figure 3.** ROC curve describing the relation of sensitivity to specificity in the clinical tests of the logistics model and MRI for the medial meniscus.

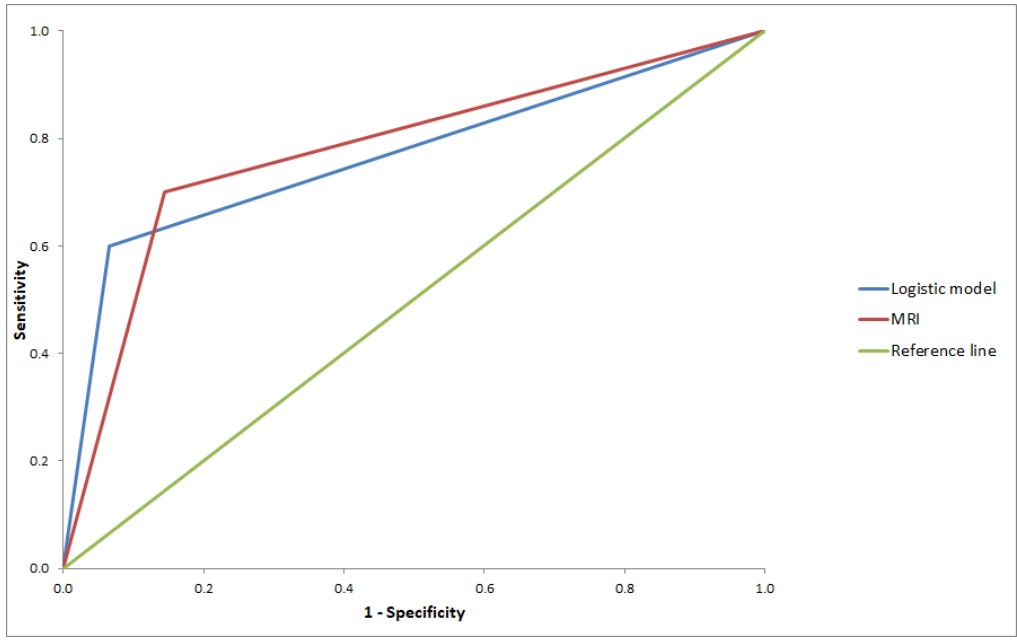

**Figure 4.** ROC curve describing the relation of sensitivity to specificity in clinical tests of the logistics model and MRI for the lateral meniscus.

*3.2. ACL Tears Diagnostic Accuracy of Physical Examination and MRI*

Out of 32 total ACL ruptures, four were not diagnosed with a physical examination. The most sensitive test for ACL tear was the Lachman test, which enabled the diagnosis of ACL tears in 27 patients. The sensitivity of the Lachman test was as high as 84%, with a high specificity of 92% as well. The anterior drawer test showed a sensitivity of 69% with a specificity of 93%. The least sensitive test in this study was the pivot shift, which showed only 43% sensitivity; however, it showed 98% specificity. Diagnostic accuracy for all tests was evaluated with regard to ROM, but no statistically significant influence was found for any of the tests. The results are shown in Table 2 and Figure 5.

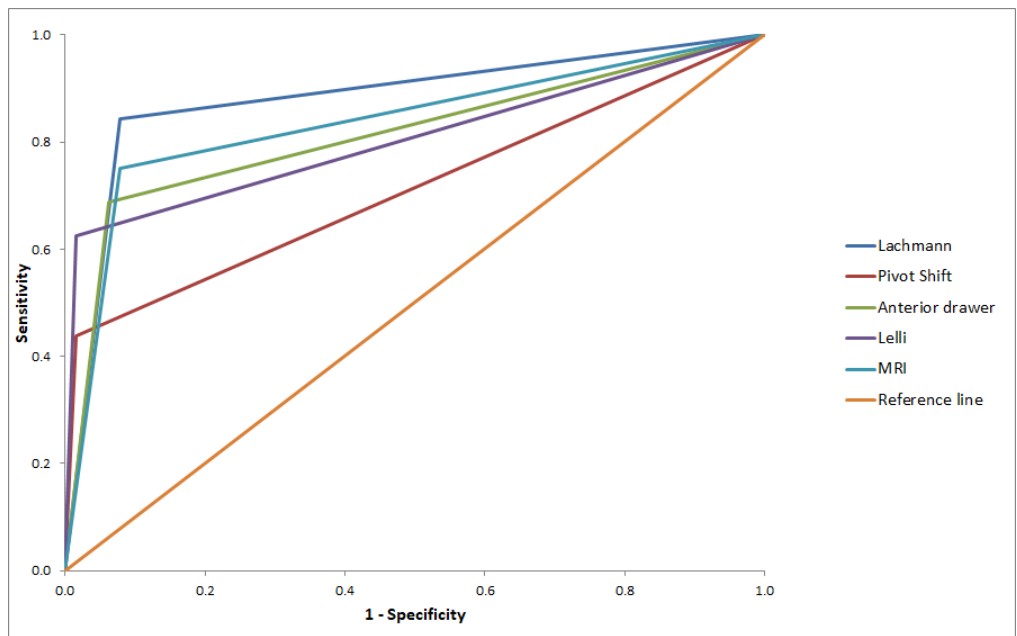

**Figure 5.** ROC curve describing the relation of sensitivity to specificity in diagnostic tests for the anterior cruciate ligament.

The sensitivity of MRI in detecting ACL ruptures was estimated to be 75% and was statistically inferior in comparison with the physical examination. There were no statistically significant differences between the specificity of the MRI and physical examination in regard to ACL ruptures.

The combined diagnostic accuracy of physical examination and MRI was also calculated. Introducing MRI into the diagnostic routine had a significant impact on diagnostic accuracy in regard to MM ($p < 0.001$) and LM ($p = 0.003$). Introducing MRI also significantly ($p < 0.001$) increased the diagnostic accuracy of the physical examination of the ACL. The results are presented in Table 2 and Figure 6.

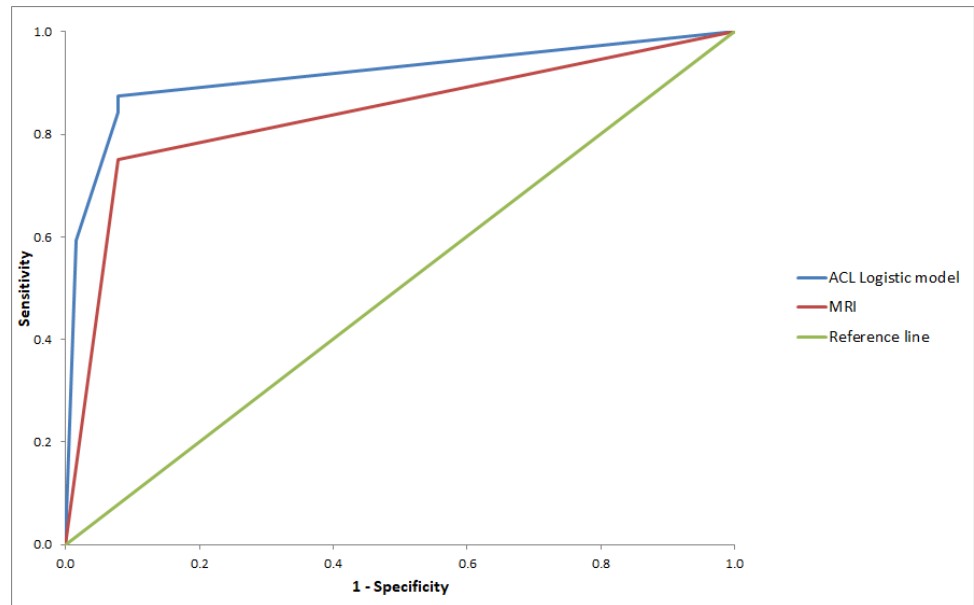

**Figure 6.** ROC curve describing the relation of sensitivity to specificity in clinical tests of the logistics model and MRI for anterior cruciate ligament.

**Table 2.** Relation of specificity to sensitivity for physical examinations and MRI for MM, LM and ACL.

| | Area Under the Curve | | | | | | |
|---|---|---|---|---|---|---|---|
| | | | | Asymptotic 95% Confidence Interval | | | |
| **Tested Variables** | **Area Under the Curve** | **SD** | **Asymptotic Significance** | **Inferior Boundary Value** | **Superior Boundary Value** | **Sensitivity** | **1-Specificity** |
| Thessaly in extension MM | 0.604 | 0.058 | 0.079 | 0.491 | 0.718 | 0.604 | 0.395 |
| Thessaly in flexion for MM | 0.698 | 0.054 | 0.001 | 0.591 | 0.805 | 0.854 | 0.458 |
| McMurray for MM | 0.698 | 0.054 | 0.001 | 0.591 | 0.805 | 0.875 | 0.479 |
| Apley compression for MM | 0.635 | 0.057 | 0.022 | 0.524 | 0.747 | 0.668 | 0.417 |
| Apley distraction for MM | 0.583 | 0.058 | 0.159 | 0.469 | 0.698 | 0.417 | 0.250 |
| Medial joint line palpation | 0.594 | 0.058 | 0.113 | 0.480 | 0.708 | 0.583 | 0.396 |
| Thessaly in extension LM | 0.746 | 0.068 | 0.001 | 0.613 | 0.879 | 0.650 | 0.158 |
| Thessaly in flexion for LM | 0.771 | 0.065 | 0.000 | 0.644 | 0.898 | 0.700 | 0.158 |
| McMurray for LM | 0.739 | 0.068 | 0.001 | 0.606 | 0.873 | 0.650 | 0.171 |
| Apley compression for LM | 0.767 | 0.070 | 0.000 | 0.630 | 0.904 | 0.600 | 0.066 |
| Apley distraction for LM | 0.705 | 0.076 | 0.005 | 0.557 | 0.853 | 0.450 | 0.039 |
| Lateral joint line palpation | 0.568 | 0.07 | 0.348 | 0.423 | 0.714 | 0.400 | 0.263 |
| Lachman | 0.883 | 0.042 | 0.000 | 0.800 | 0.966 | 0.844 | 0.078 |
| Pivot shift | 0.711 | 0.062 | 0.001 | 0.589 | 0.832 | 0.438 | 0.016 |
| Anterior drawer | 0.813 | 0.053 | 0.000 | 0.709 | 0.916 | 0.688 | 0.063 |
| Lelli | 0.805 | 0.055 | 0.000 | 0.696 | 0.913 | 0.625 | 0.016 |
| MRI for MM | 0.740 | 0.052 | 0.000 | 0.638 | 0.841 | 0.958 | 0.479 |
| Logistic model for MM | 0.726 | 0.053 | 0.000 | 0.623 | 0.830 | 0.917 | 0.542 |
| MRI for LM | 0.778 | 0.065 | 0.000 | 0.650 | 0.905 | 0.7 | 0.145 |
| Logistic model for LM | 0.767 | 0.070 | 0.000 | 0.630 | 0.904 | 0.6 | 0.066 |
| MRI for ACL | 0.836 | 0.050 | 0.000 | 0.739 | 0.933 | 0.75 | 0.078 |
| Logistic model for ACL | 0.914 | 0.050 | 0.000 | 0.739 | 0.933 | 0.875 | 0.078 |

## 4. Discussion

Knee joint disfunction is one of the most common problems in daily orthopaedic practice. It may be caused by previous injuries, congenital disorders or other conditions acquired during living, which, untreated, may lead to irreversible osteoarthritic changes of the knee joint.

As in the published literature [22–24], the most common procedures in our group were meniscal procedures. However, it is well known that partial meniscectomy accelerates cartilage degeneration. The first line of treatment in older populations with meniscal lesions and coexisting cartilage defects is conservative treatment [24]. This may explain the prolonged time from the onset of symptoms to surgical treatment in our study, since most of the patients had undergone previous conservative treatment. However the mean visual analogue scale (VAS) score was 5.4, which can lead to great dysfunction in daily activities; therefore, it seems that such a long period of awaiting surgical treatment (44 months) may be too long for patients to bear.

In our study, we found that the McMurray and Thessaly tests were the most sensitive in detecting medial meniscus lesions, with 85–87% sensitivity. Even though McMurray [13] in his original work described a clunk during knee motion, over the years various modifications have been introduced to the test, including the onset of pain or adding valgus and varus force to the knee. In our study we have shown a higher sensitivity of the McMurray test in detecting medial meniscus lesions than in the previously published papers [25,26]. Such a high sensitivity in our results may be an effect of the preference of the examining surgeon for performing the McMurray test; however, the previously published papers showed that experience in performing this particular test plays no role in the diagnostic accuracy of the test [25], and therefore it seems an appropriate test to incorporate in the routine examination of patients. Most of the published literature has shown that the sensitivity of the McMurray test is lower for the lateral meniscus than for the medial meniscus [25,27,28]. Our results confirm this finding; however, the paper published by Karachalios et al. suggests that the sensitivity of the McMurray test is increased for the lateral meniscus [15].

In 2005 a novel diagnostic test for meniscal tears was introduced [15] that was suggested to show higher sensitivity and specificity. According to Karachalios et al. [15], a higher sensitivity in detecting lateral meniscus tears has been one of the most important findings of that test. However, our results and data published previously in literature oppose these findings [29], showing in fact a decreased sensitivity of the Thessaly test for the lateral meniscus. This divergence might be due to the fact that the Thessaly test is a dynamic examination in which the patient is actively involved and has to perform several rapid movements of the limb while bearing weight. In our study, we found a significant dependence of the diagnostic accuracy of the test in relation to the extension lag of the knee joint ($p = 0.012$). The Thessaly test was also the only test showing a dependence on prior physiotherapy ($p = 0.005$). These findings may contribute to understanding the gross divergence of data related to the Thessaly test.

The Apley test was primarily designed in order to distinguish meniscal lesions from other intraarticular pathologies. However, our results differed from other publications in regard to sensitivity and specificity of the Apley test, finding both to be lower than in the previously published papers [15,28]. In a paper published by Speziali et al. [28], the exclusion criteria included osteoarthritic changes in the joint. In our study we tried to answer the question of whether chondral lesions influence the diagnostic accuracy of physical examinations of meniscal lesions. We found that all of the tests were influenced by osteoarthritic changes in the knee joint with increasing sensitivity and decreasing specificity in higher-grade cartilage defects. Based on European Society for Sports Traumatology, Knee Surgery and Arthroscopy (ESSKA) [24] recommendations it seems highly important to distinguish between pain caused by OA and meniscal lesions, because this changes the therapeutic route. Therefore we believe that this test should be incorporated in routine knee checkups.

In our study we have also tested the diagnostic accuracy of the ACL deficiency test. We found the Lachman test to be the most sensitive in this regard, with high specificity in detecting ACL laxity. Our results correspond with the literature on this subject. A meta-analysis published by

Benjaminse et al. [30] of 28 articles showed, similar to our results, a sensitivity of 85% and specificity of 94% for the Lachman test. However, papers with a reported sensitivity lower than 50% can also be found in the literature [31,32]. In these articles, examination of the knee was performed in an acute stage after knee injury, and therefore hemarthrosis or post-injury pain might have affected the clinical findings. As mentioned above, the Lachman examination might have been hampered by knee pain or hemarthrosis, and for this reason many orthopaedic surgeons use the anterior drawer test in testing anterior laxity of the knee joint. In our study, the sensitivity of the anterior drawer test was found to be lower than 70%, which corresponded with recent literature on the subject [33]. Benjaminse et al. [30] even suggest that the sensitivity of the test could be lower than 50%. The authors suggest that such low sensitivity might be caused by hemarthrosis, inadequate relaxation of the patient during examination or locking of the knee joint by a torn meniscus. In our study we examined patients one day prior to surgery, which was on average 44 months after the onset of symptoms. Therefore, the bias of pain and discomfort during acute setting was not present at the time of examination.

Rotational instability is the main patient-disturbing symptom of ACL deficiency [34,35]. The commonly used pivot shift (PS) test was designed in order to detect rotational instability in the knee joint. Nevertheless, in our study we have found PS to be extraordinarily unsensitive in detecting ACL deficiency, although PS showed the highest specificity among all the tested examinations. Other studies have shown similar results in regard to PS examination [30,33]. The sensitivity of the test is reported to grossly increase when the patient is sedated or under general anesthesia, with a sensitivity rate of 85% [31] due to the great discomfort of patients during the examination and their reactive muscle responses. To the best of our knowledge, our study is the fourth to evaluate Lelli's lever sign for ACL deficiency. In his original work, Lelli [19] reported 100% sensitivity of his test even in an acute setting. In our study we found the sensitivity of the test to be 62.5%, with a specificity of 98% as in PS. Two other studies that can be found in literature on the subject also did not report such a high sensitivity. Massey et al. [36] estimated the sensitivity of the test to be 83%, and the authors suggest that such a high diagnostic value reported in the original paper might have been caused by the methodology of the study in which the author verified his test on patients with known ACL deficiency, which could have affected the final result. Furthermore, in a study published by Jarbo et al. [37], high sensitivity was again not confirmed. What is worth noting is the fact that this examination, contrary to others tested in the study, was independent of patient relaxation, and therefore it might be utilized in daily practice in an acute setting.

MRI is considered to be the most accurate method of imaging of the internal knee joint structure, with sensitivity in detecting medial meniscus lesions ranging from 83% to 94% [38,39]. Our study have showed similar sensitivity with regard to medial meniscus lesions reaching 95.6%. However, some studies suggest a lower sensitivity of the MRI in detecting medial meniscus lesions. Yoon et al. [40] evaluated the sensitivity to be 74%. However, in mentioned paper, the authors studied acute knee injuries with hemarthrosis, which could have influenced the results. Our results also correspond with the literature regarding the sensitivity and specificity of MRI for lateral meniscus lesions. Phalen et al. [38], in their meta-analysis, estimated a sensitivity ranging from 66% to 87%. In our study, we found 70% sensitivity with a specificity of 85%. It is well demonstrated in literature that MRI has a lower sensitivity in detecting lateral meniscus lesions, which some authors attribute to its anatomical structure [41]. Moreover, the location of a lesion impacts the diagnostic accuracy of MRI. Both in the published papers [42] and in our study, the posterior parts of the meniscus have been more accurately diagnosed than the anterior regions.

In our study we found MRI to have a 75% sensitivity in detecting ACL rupture. In the available literature, the evaluated sensitivity was not reported lower than 77%, even though some studies have been conducted on 0.2T coils [43,44]. Phelan et al. [38] brought attention to a possible bias of the studies due to their founding by coil manufacturers. Nevertheless, it does not alter the fact that in our study the sensitivity was lower than has been previously reported in other papers. In 2012, van Dyck et al. [45] evaluated the phenomenon of partial ACL tears found in MRI and their implication

on surgery planning. The authors concluded that the evaluation of partial ACL tears is unreliable in that partial tears, and even the usage of 3T coils, do not improve results [46]. Moreover, the MRI examinations in our study were made in different departments and we were not able to retrieve the referrals of all examinations to verify if the referral included all important information for the leading radiologist. However, this situation resembled a typical clinical setting and the results might be more helpful in every day practice. This data suggest that clinical examinations are grossly superior to the MRI in detecting ACL deficiency, and that MRI has limited application in detecting isolated ACL tears.

## 5. Conclusions

The McMurray test was found to be the most sensitive test for diagnosing medial meniscus lesions, whilst the Thessaly test was the most sensitive for the lateral meniscus. The Apley test was the most specific for both the medial and the lateral meniscus. Therefore, these two tests should be implemented in a daily routine examining knee joints. Joint line palpation was found to be the least sensitive test for the medial and the lateral meniscus. Regarding ACL, the Lachman test showed the best sensitivity, while the lever sign and pivot shift were shown to be the most specific tests for ACL deficiency. As a result, a combination of these tests seems most applicable for detecting ACL insufficiency.

MRI is more accurate in diagnosing medial and lateral meniscus lesions than physical examination; however, the difference is not statistically significant and MRI should be used as an additional examination to evaluate the extent of meniscal lesion rather than just plain diagnosis. Nevertheless, MRI shows lower accuracy in detecting ACL deficiency than physical examination, a finding that is statistically significant.

Patients should be referred to MRI to answer specific questions about the morphology of lesions prior to surgery, rather than to confirm diagnosis, which can be established by proper physical examination.

**Author Contributions:** Conceptualization, methodology, investigation, writing—original draft preparation, resources, P.K.; supervision, A.N. and A.J.; formal analisys, J.J. and R.M.; writing—review & editing, validation, R.K.

**Funding:** The research was financed in the framework of the project Lublin University of Technology—Regional Excellence Initiative, funded by the Polish Ministry of Science and Higher Education (contract no. 030/RID/2018/19).

**Conflicts of Interest:** The authors declare no conflict of interest.

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
