# Peer review of "Comparison of Diagnostic Accuracy of Physical Examination and MRI in the Most Common Knee Injuries"

_applsci, doi:10.3390/app9194102_

Round 1

Reviewer 1 Report

              The study is very interesting and it provides very interesting information, however it needs major corrections. I would like to emphasize the effort that had to be put in the preparation of this research.

              Firstly, remember that the Journal is not a standard orthopaedic one. The manuscript has to be rewritten so it was understood to anybody – not only an orthopaedic surgeon.

              Secondly, I think that if the physical examination was focused on ACL and menisci injuries diagnosis, if the authors want to compare the physical examination to MRI, then the cartilage lesions should be excluded from the study. I would leave ACL and menisci, and I would create another manuscript where I would compare MRI with arthroscopy findings in cartilage lesions diagnostic. I think it would make the manuscript and its findings more clear. Sometimes, less is more.

              As  I understand the purpose of the study was to correlate specific clinical tests and MRI evaluation of ACL and menisci injuries. Generally, this correlation should be defined by calculating sensitivity, specificity, positive and negative predictive values (PPV & NPV) keeping arthroscopy as a gold standard. Please refer to it and explain your approach to the problem. It is worth to add a subsection in the Materials and Methods titled ‘Experimental Approach to the Problem’, because the idea seems a bit unclear. Once you decide what you want to discuss, please rewrite the manuscript according to it.

1.      The title

Firstly, I would always use the ‘physical examination’ first, instead of ‘MRI’ as generally it’s the primary examination. So in the title as well as in the aim of the study, conclusions etc. it should be used as a first one. I also think it should be written in ‘the most common’ knee joint injuries, however, I’m not a native speaker. Please do not use point after the title.

2.      Abstract

Line 18: In the Purpose of the study please write ‘in the most common knee joint injuries’ instead of ‘disorders’.

Line 20: Methods; when we start a sentence in with the number, we shouldn’t write it as numbers, so the sentence should start ‘Ninety-six patients’.

Line 20: Do not write details about hospital in the abstract. Write for example ‘The initial sample consisted of x patients referred to the hospital (academic, provincial, regional?) for the knee joint. The final sample consisted of 96 patients who underwent knee joint physical examination of the knee joint and whose results of the knee MRI were analysed. Then a diagnostic knee arthroscopy was performed. The physical examination included clinical tests for ACL insufficiency (pivot-shift, Lachman, anterior drawer, lever sign) and meniscus (joint line palpation, Thessaly, Apley, McMurray). The physical examination findings were compared to MRI results keeping arthroscopy as a gold standard.’

Line 24: Add a sentence about data analysis.

Line 24: The results section is definitely too long. In the abstract show only the most important results – those results that would determine the conclusions and that are strictly in line with the purpose of the study.

Line 35: The conclusions that MRI is helpful and that the utility of MRI is limited are not really innovative. Your study has far more bigger potential that you have showed in the conclusions. Please answer for the question how the clinical tests you used correlates with the MRI findings as this is the most important part of your study. As you wrote in the Introduction – you wanted to create an algorithm which would enable orthopaedic surgeon to effectively and fast diagnose the knee injury and decide about the treatments. You are armed with the studied material that can create this algorithm, however, it seems like you don’t know how to describe it.

3.      Introduction

The introduction is far too long. In fact, the most important information starts from the line 70. I guess you should rewrite the Introduction. Start with emphasizing the frequency and consequences of ACL and menisci injuries. Then write about the standard diagnosis of those injuries. The paragraph saying about overuse of MRI is very interesting and for me it’s a good support the purpose of your study. Also the idea of creating an algorithm is very good, however, your results, discussion and conclusions have to be in line with your idea.

Please rewrite the purpose of the study.

4.      Materials and Methods

I would recommend you to start with an Experimental Approach to the Problem (as above).

Please add subsection Ethics – you have missed those information. For example:

The study was carried out in the Orthopaedic Department of Łęczna Hospital in Poland in years …-…. The study was conducted according to the ethics guidelines and principles of the Declaration of Helsinki The study was approved by the (give the name and the number of ethics committee ). All of the studied participants were informed about the goal of the study and approached to be used, and signed their informed consent.

Please define the study design. Once you define the study design, use an appropriate clinical research checklist tool (for example CONSORT or other depending on your study design) to prepare the manuscript.

Please add the subsection ‘Participants’. Start with defining the initial sample and then provide inclusion and exclusion criteria, and finally introduce the final sample. I have notice that a large part of the Results section should be replaced into the Participants subsection. As long as your study is not an epidemiological one, the data concerning age, sex distribution, duration of symptoms etc. belongs to characteristic of the studied sample. In the Results section there is a Table 1 where you have included the percentage distribution of different injuries. It is not clear whether does are injuries diagnosed in during the physical examination, MRI or arthroscopy (????). Please put the right data into the right section where they belong to. Provide the flow diagram of the study.

At this point, I would like to draw your attention to the very important question that arose during the reading of your work. Is it possible that too long time (actually what time?) between MRI and physical examination could have influenced the results of comparative analysis of these two types of examinations? Did you take this into account? Was there any defined time between MRI and physical examination prior to the study? Please consider this very seriously, because too disrespectful approach to the problem could undermine all your conclusions.

After the ‘Participants’ subsection please provide following subsections: ‘Physical Examination of the Knee Joint’, ‘Magnetic Resonance Imaging of the Knee Joint’, ‘Knee Arthroscopy’, ‘Data Analysis’, ‘Statistical Analysis’, and describe each subsection in detail. Please remember, that not all of the readers are orthopaedic surgeons. Don’t miss the information concerning the number of examiners in particular parts of the examination, blinding the examiners, time between the MRI and physical examination, time between the MRI and arthroscopy, time between the physical examination and arthroscopy etc. Please remember that this subsection should in principle be written with the utmost care, and in your case it was described most briefly in the whole manuscript. this chapter is the core of the whole work, so it should be as clear as possible.

5.      Results

As I have mentioned before, this section needs serious revision. Firstly, the characteristic of the studied sample should be replaced to the ‘Participants’ section, and secondly, include only the results that are in line with the purpose of the study! The graphs are very unclear and unfriendly to the reader.

6.      Discussion

It should be rewritten. I think you should focus on the algorithm you mentioned in the Introduction and it should be based on your experience from this study, and the studies of other authors. Don’t forget about the limitations of the study (different MRI machines etc.) and provide practical applications of your study.

7.      Conclusions

Please rewrite the Conclusions so they were in line with the Purpose of the study.

Author Response

            The study is very interesting and it provides very interesting information, however it needs major corrections. I would like to emphasize the effort that had to be put in the preparation of this research.

1)      Firstly, remember that the Journal is not a standard orthopaedic one. The manuscript has to be rewritten so it was understood to anybody – not only an orthopaedic surgeon.

-In order to answer the question whether the manuscript is understandable for non-ortoppaedic surgeons we have asked couple of physician and engineers to read the manuscript. There were no complaints about comprehensibility of the article.

2)    Secondly, I think that if the physical examination was focused on ACL and menisci injuries diagnosis, if the authors want to compare the physical examination to MRI, then the cartilage lesions should be excluded from the study. I would leave ACL and menisci, and I would create another manuscript where I would compare MRI with arthroscopy findings in cartilage lesions diagnostic. I think it would make the manuscript and its findings more clear. Sometimes, less is more.

       -There are no specific tests that can be used in detection of cartilage lesions. Therefore the only way to diagnose cartilage defects is MRI. We believe that omitting diagnostic accuracy of MRI in cartilage lesions would lessen the value of the manuscript by omitting the most common finding in the study.

3)    As  I understand the purpose of the study was to correlate specific clinical tests and MRI evaluation of ACL and menisci injuries. Generally, this correlation should be defined by calculating sensitivity, specificity, positive and negative predictive values (PPV & NPV) keeping arthroscopy as a gold standard. Please refer to it and explain your approach to the problem. It is worth to add a subsection in the Materials and Methods titled ‘Experimental Approach to the Problem’, because the idea seems a bit unclear. Once you decide what you want to discuss, please rewrite the manuscript according to it.

       -In this manuscript we have decided to show results in a graphical way presented in ROC curves. The ROC curves are created by correlation of true positives, negatives and false positives and negatives. All these data was calculated during preparation of the manuscript, however in order not to lengthen the article, these data was omitted and only graphical presentation was shown.

4)      The title Firstly, I would always use the ‘physical examination’ first, instead of ‘MRI’ as generally it’s the primary examination. So in the title as well as in the aim of the study, conclusions etc. it should be used as a first one. I also think it should be written in ‘the most common’ knee joint injuries, however, I’m not a native speaker. Please do not use point after the title.

-Accordingly to the reviewer’s suggestions we have changed the title of the manuscript.

5)      Abstract

Line 18: In the Purpose of the study please write ‘in the most common knee joint injuries’ instead of ‘disorders’.

Line 20: Methods; when we start a sentence in with the number, we shouldn’t write it as numbers, so the sentence should start ‘Ninety-six patients’.-

Line 20: Do not write details about hospital in the abstract. Write for example ‘The initial sample consisted of x patients referred to the hospital (academic, provincial, regional?) for the knee joint. The final sample consisted of 96 patients who underwent knee joint physical examination of the knee joint and whose results of the knee MRI were analysed. Then a diagnostic knee arthroscopy was performed. The physical examination included clinical tests for ACL insufficiency (pivot-shift, Lachman, anterior drawer, lever sign) and meniscus (joint line palpation, Thessaly, Apley, McMurray). The physical examination findings were compared to MRI results keeping arthroscopy as a gold standard.’

Line 24: Add a sentence about data analysis.

Line 24: The results section is definitely too long. In the abstract show only the most important results – those results that would determine the conclusions and that are strictly in line with the purpose of the study. It has been changed accordingly to review.  

Line 35: The conclusions that MRI is helpful and that the utility of MRI is limited are not really innovative. Your study has far more bigger potential that you have showed in the conclusions. Please answer for the question how the clinical tests you used correlates with the MRI findings as this is the most important part of your study. As you wrote in the Introduction – you wanted to create an algorithm which would enable orthopaedic surgeon to effectively and fast diagnose the knee injury and decide about the treatments. You are armed with the studied material that can create this algorithm, however, it seems like you don’t know how to describe it.

All sections needing rewriting were checked and rewritten.

The purpose of this study was to evaluate the diagnostic accuracy of MRI and physical examination which could help in creating a diagnostic algorithm, however development of the algorithm was not a study goal itself. However, suggestions of tests that should be incorporated in daily practice were made in the text.

6)       Introduction

The introduction is far too long. In fact, the most important information starts from the line 70. I guess you should rewrite the Introduction. Start with emphasizing the frequency and consequences of ACL and menisci injuries. Then write about the standard diagnosis of those injuries. The paragraph saying about overuse of MRI is very interesting and for me it’s a good support the purpose of your study. Also the idea of creating an algorithm is very good, however, your results, discussion and conclusions have to be in line with your idea. The introduction was shortened and consolidated, with emphasis put on timing of the diagnosis as well as the articular cartilage, while it is the single most important tissue of every synovial joint and all other tissues serves only the purpose of protecting articular cartilage.

Please rewrite the purpose of the study. The last paragraph of introduction including aim of the study was rewritten accordingly to the suggestions.

7)        Materials and Methods

I would recommend you to start with an Experimental Approach to the Problem (as above).

Please add subsection Ethics – you have missed those information. For example:

The study was carried out in the Orthopaedic Department of Łęczna Hospital in Poland in years …-…. The study was conducted according to the ethics guidelines and principles of the Declaration of Helsinki The study was approved by the (give the name and the number of ethics committee ). All of the studied participants were informed about the goal of the study and approached to be used, and signed their informed consent.

 The information about written consent of the patients was added to the text.  

Please define the study design. Once you define the study design, use an appropriate clinical research checklist tool (for example CONSORT or other depending on your study design) to prepare the manuscript.

The study was a prospective study and the information about the study was added in the text of this section.  

Please add the subsection ‘Participants’. Start with defining the initial sample and then provide inclusion and exclusion criteria, and finally introduce the final sample. I have notice that a large part of the Results section should be replaced into the Participants subsection. As long as your study is not an epidemiological one, the data concerning age, sex distribution, duration of symptoms etc. belongs to characteristic of the studied sample. In the Results section there is a Table 1 where you have included the percentage distribution of different injuries. It is not clear whether does are injuries diagnosed in during the physical examination, MRI or arthroscopy (????). Please put the right data into the right section where they belong to. Provide the flow diagram of the study.

The section “participants has been added accordingly to the suggestions. The table 1 refers to the diagnosis based on arthroscopic findings, however it might be a little bit confusing, therefore additional information in the text was added.

At this point, I would like to draw your attention to the very important question that arose during the reading of your work. Is it possible that too long time (actually what time?) between MRI and physical examination could have influenced the results of comparative analysis of these two types of examinations? Did you take this into account? Was there any defined time between MRI and physical examination prior to the study? Please consider this very seriously, because too disrespectful approach to the problem could undermine all your conclusions.

The comment is very important. The time between MRI and arthroscopy was calculated and mean time was 3,6 months with SD 3,3. Therefore it represents actual time that patients await in for treatment and it should not impede the diagnostic accuracy of MRI

After the ‘Participants’ subsection please provide following subsections: ‘Physical Examination of the Knee Joint’, ‘Magnetic Resonance Imaging of the Knee Joint’, ‘Knee Arthroscopy’, ‘Data Analysis’, ‘Statistical Analysis’, and describe each subsection in detail. Please remember, that not all of the readers are orthopaedic surgeons. Don’t miss the information concerning the number of examiners in particular parts of the examination, blinding the examiners, time between the MRI and physical examination, time between the MRI and arthroscopy, time between the physical examination and arthroscopy etc. Please remember that this subsection should in principle be written with the utmost care, and in your case it was described most briefly in the whole manuscript. this chapter is the core of the whole work, so it should be as clear as possible.

 The materials and methods was divided into clear subsections. Additional information about data collection and medical personnel performing each part was added to the text to clarify the methodology of the study.

8)        Results

As I have mentioned before, this section needs serious revision. Firstly, the characteristic of the studied sample should be replaced to the ‘Participants’ section, and secondly, include only the results that are in line with the purpose of the study! The graphs are very unclear and unfriendly to the reader. 

Unnecessary data was excluded from this section. Graphs were redone to make them more friendlier for the readers.

9)        Discussion

It should be rewritten. I think you should focus on the algorithm you mentioned in the Introduction and it should be based on your experience from this study, and the studies of other authors. Don’t forget about the limitations of the study (different MRI machines etc.) and provide practical applications of your study.

The discussion was, shortened and rewritten. The emphasis on proposed tests which should be used in daily practice was added in the discussion.

10)    Conclusions

Please rewrite the Conclusions so they were in line with the Purpose of the study.  The conclusions were rewritten in order to harmonize with the aim of the study.

Reviewer 2 Report

Excellent presentation of nice paper. Some suggested conclusions will be very helpful in routine work.

Author Response

No comments requiring reply. 

Reviewer 3 Report

Lelli test has a sensitivity (the percentage of ACL rupture who are correctly identified) of 100% if properly performed. If the fist of the examinator is not positioned in the right place it could bring to false negative

Author Response

Lelli test has a sensitivity (the percentage of ACL rupture who are correctly identified) of 100% if properly performed. If the fist of the examinator is not positioned in the right place it could bring to false negative

In our study Lelli test showed specificity of almost 100%, however the sensitivity of the test was rather low, which corresponds with papers published on the subject. Maybe this test needs evaluation on greater scale in order to determine its clinical utility.

Round 2

Reviewer 1 Report

If the authors decided to leave the patients with chondral lesions who didn’t undergo the physical examination, then the title and the aim of the study doesn’t match the methodology of the study and the whole  methodology of the study doesn’t make sense at all. I really do know that there are no specific tests for chondral lesions. That’s why it make no sense from the methodological point of view. If there is a group that didn’t have physical examination you can’t write that you are comparing the diagnostic accuracy of physical examination and MRI.

Abstract

Line 25: In the abstract the authors use MRI, while they explain the short name in the Introduction writing “magnetic resonance imaging in the line 55. It doesn’t make any sense.

Line 26: Regional instead of regoinal.

Line 27: Should be physical (not clinical) examination of the knee joint or knee joint physical examination.

Line 27: There should be knee joint (or knee) MRI.

Line 27: As I understand, the authors have evaluated only one ligament – the ACL, not “ligaments”.

Line 28: There should be knee arthroscopy.

Line 29: Please explain ROC curves. On the other hand, the abstract has to be written as a separate text. That means that if you don’t show ROC curves in the abstract, then don’t write about it. The most important thing is what do the percentage values mean, because they are not explained and not something that is within the manuscript.

Line: 31: “The most common findings were…” – Where were those findings? In the arthroscopy?

Line 32: It should be medial meniscus instead of Medial meniscus.

Line 33: The authors wrote anterior cruciate ligament, even though the short “ACL” is used in line 31.

Line 34: What is the Lelli test? It hasn’t been mentioned in the methodology.

The authors should order the manuscripts in terms of the order of the described structures of the knee joint. I mean, if in the first place they write about meniscus and then about ligaments (ligament?) then they should stick to it throughout the whole manuscript.

Introduction

Line 63: This description is too detailed.

Line 76-78: This should be replaced.

Materials and Methods

The authors still haven’t provided any information concerning the name of the ethical committee that had approved the study.

Line 82: The sentence doesn’t make sense as the patients have to be informed about the goal of the study and approach to be used before they sign an informed written consent.

Remember about the chronology when you write about participants:

Provide information about the number of patients in the initial sample.

Provide exclusion criteria and determine the number of patients excluded for particular reasons.

Characterise the group that was left. You can write that the detailed characteristic of studied sample for presented in Table 1 and present data concerning sex, age etc.

I don’t know why the information concerning VAS results was placed in the characteristic of studied material. If it was a part of  the examination protocol, then it should be moved to the next subsection. However, I don’t understand the aim of providing the pain intensity assessment if the aim of the study was about something else.

Data and statistical analysis

This subsection is far too long and it doesn’t provide any specific information.

When you write about the program that was used, you should write the name of the producent, year, city, country etc.

Bulleting is doubled.

Line 166: Something is wrong as the Shapiro-Wilk test provides information concerning the distribution. So we don’t make it in case of normal distribution but we know that the distribution is normal because we have done this test.

You have to be more specific – don’t write “for continuous variables” – but write for which variables specifically. There weren’t many variables that you have analysed so just name them.  The same with “two-tier variables” (?) etc.

Conclusions

The conclusions are far too long.

Author Response

If the authors decided to leave the patients with chondral lesions who didn’t undergo the physical examination, then the title and the aim of the study doesn’t match the methodology of the study and the whole  methodology of the study doesn’t make sense at all. I really do know that there are no specific tests for chondral lesions. That’s why it make no sense from the methodological point of view. If there is a group that didn’t have physical examination you can’t write that you are comparing the diagnostic accuracy of physical examination and MRI.

All patients underwent physical examination, however there are no specific tests for chondral lesions detection as noticed by the reviewer. The only way to detect chondral lesions is MRI, therefore it seemed important for us to evaluate whole spectrum of MRI utility in most common knee injuries. However, after consideration the arguments raised by reviewer were implemented in the manuscript and chondral lesions were excluded from the manuscript which made the paper more coherent for the readers.

Abstract

Line 25: In the abstract the authors use MRI, while they explain the short name in the Introduction writing “magnetic resonance imaging in the line 55. It doesn’t make any sense.

It was corrected

Line 26: Regional instead of regoinal.

The spelling mistake was corrected

Line 27: Should be physical (not clinical) examination of the knee joint or knee joint physical examination.

The nomenclature was corrected

Line 27: There should be knee joint (or knee) MRI.

The sentence was changed

Line 27: As I understand, the authors have evaluated only one ligament – the ACL, not “ligaments”.

All ligaments were evaluated, however there was only one patient in the group with PCL deficiency, and LCL, MCL, MPFL are extraarticular ligaments and can’t be checked during arthroscopy, therefore only results for ACL were included in the study and mentioned in the abstract as well as in the text of the manuscript. However, we find the reviewers comment important to specify the abstract for ACL. It was corrected.

Line 28: There should be knee arthroscopy.

It was corrected

Line 29: Please explain ROC curves. On the other hand, the abstract has to be written as a separate text. That means that if you don’t show ROC curves in the abstract, then don’t write about it. The most important thing is what do the percentage values mean, because they are not explained and not something that is within the manuscript.

The explanation was added, that sensitivity and specificity were calculated and then correlated on ROC curves.

Line: 31: “The most common findings were…” – Where were those findings? In the arthroscopy?

It was rewritten accordingly to reviewers suggestions.

Line 32: It should be medial meniscus instead of Medial meniscus.

The spelling error was corrected.

Line 33: The authors wrote anterior cruciate ligament, even though the short “ACL” is used in line 31.

It was corrected.

Line 34: What is the Lelli test? It hasn’t been mentioned in the methodology.

None of the tests used in the study was mentioned in the methodology in abstract because it would lengthen the abstract to much. But all tests were included in results section of the abstract.

The authors should order the manuscripts in terms of the order of the described structures of the knee joint. I mean, if in the first place they write about meniscus and then about ligaments (ligament?) then they should stick to it throughout the whole manuscript.

The manuscript was organised in a manner that menisci are mentioned in each subsection prior to ACL.

Introduction

Line 63: This description is too detailed.

The description was shortened.

Line 76-78: This should be replaced.

This was rewritten.

Materials and Methods

The authors still haven’t provided any information concerning the name of the ethical committee that had approved the study.

Line 82: The sentence doesn’t make sense as the patients have to be informed about the goal of the study and approach to be used before they sign an informed written consent.

All patients were informed that the results of their examination without revealing any personal data will be used in a study. Each patient signed a consent for the use of results of their examinations in standard Hospital policy. The patients who had not signed the consent were not included in the study. Since the study was performed only for evaluation of clinical tests in patients previously scheduled for surgery, there was no additional risk of participating in the study. The physical examination, collection of imaging studies and recording of surgical procedures is a standard protocol, therefore no additional Committee approval was required, but all necessary precautions stated in The Hospital policy of protecting personal data were implemented.

Remember about the chronology when you write about participants:

Provide information about the number of patients in the initial sample.

Provide exclusion criteria and determine the number of patients excluded for particular reasons.

The exclusion criteria were given in participants section. However, since the exclusion from the study had no influence on treatment regimen, the exact count of the excluded patients was not obtained and there is no record of that.

Characterise the group that was left. You can write that the detailed characteristic of studied sample for presented in Table 1 and present data concerning sex, age etc.

I don’t know why the information concerning VAS results was placed in the characteristic of studied material. If it was a part of  the examination protocol, then it should be moved to the next subsection. However, I don’t understand the aim of providing the pain intensity assessment if the aim of the study was about something else.

The VAS score was evaluated in order to crosscheck if it can influence the physical examination. There have been found no correlation between VAS and physical examination so it was omitted in the text, however we believed that simple information about severity of pain which brings patients to seek medical assistance could provide better understanding of the treated group. Nevertheless, accordingly to reviewers suggestions this sentence was removed from the manuscript.

Data and statistical analysis

This subsection is far too long and it doesn’t provide any specific information.

The description was shortened.

When you write about the program that was used, you should write the name of the producent, year, city, country etc.

Bulleting is doubled.

It was corrected accordingly to reviewers suggestions.

Line 166: Something is wrong as the Shapiro-Wilk test provides information concerning the distribution. So we don’t make it in case of normal distribution but we know that the distribution is normal because we have done this test.

Additional information was provided on the statistical methods used to check whether the values of the tested variables are distributed in accordance with normal distribution.

You have to be more specific – don’t write “for continuous variables” – but write for which variables specifically. There weren’t many variables that you have analysed so just name them.  The same with “two-tier variables” (?) etc.

The tests were assigned to specific variables and tests that were not included in revised version of manuscript were removed.

Conclusions

The conclusions are far too long.

The conclusions were shortened and present only the most important findings of the study.

Round 3

Reviewer 1 Report

Because one of the review points is to verify that the authors of the work have obtained the consent of the ethics committee, I must address the question to the editors if in this particular manuscript the consent of the ethics committee is needed. The authors in their comments to the reviewer have written that they did not need this consent, quoting  "Since the study was performed only for evaluation of clinical tests in patients previously scheduled for surgery, there was no additional risk of participating in the study. The physical examination, collection of imaging studies and recording of surgical procedures is a standard protocol, therefore no additional Committee approval was required (...) ". I kindly ask the editors of the journal to respond to this issue as the study had prospective design.

Author Response

Because one of the review points is to verify that the authors of the work have obtained the consent of the ethics committee, I must address the question to the editors if in this particular manuscript the consent of the ethics committee is needed. The authors in their comments to the reviewer have written that they did not need this consent, quoting  "Since the study was performed only for evaluation of clinical tests in patients previously scheduled for surgery, there was no additional risk of participating in the study. The physical examination, collection of imaging studies and recording of surgical procedures is a standard protocol, therefore no additional Committee approval was required (...) ". I kindly ask the editors of the journal to respond to this issue as the study had prospective design.

The reply to the editors suggestions was made and sent.